# Theoretical study of deeply virtual Compton scattering off $^4$He

Sara Fucini[1] $^\star$, Sergio Scopetta[1] and Michele Viviani[2]

**1** Dipartimento di Fisica e Geologia, Università degli Studi di Perugia and Istituto Nazionale di Fisica Nucleare, Sezione di Perugia, via A. Pascoli, I - 06123 Perugia, Italy
**2** INFN-Pisa, 56127 Pisa, Italy
$^\star$ sara.fucini@pg.infn.it

October 15, 2019

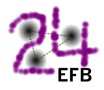

*Proceedings for the 24th edition of European Few Body Conference, Surrey, UK, 2-6 September 2019*

## Abstract

**An interesting breakthrough in understanding the elusory inner content of nuclear systems in terms of partonic degrees of freedom is represented by deeply virtual Compton scattering processes. In such a way, tomographic view of nuclei and bound nucleons in coordinate space could be achieved for the first time. Moreover, nowadays experimental results for such a process considering $^4$He targets recently released at Jefferson Lab are available. In this talk, the recent results of our rigorous Impulse Approximation for DVCS off $^4$He, in terms of state-of-the-art models of the nuclear spectral function and of the parton structure of the bound proton, able to explain present data, has been shown.**

## 1 Introduction

Recently it has become clear that inclusive Deep Inelastic Scattering measurements do not allow to fully understand the elusive parton structure of nuclei and nucleons. In facts, the

quantitative understanding of the origin of the EMC effect [1], i.e. the nuclear medium modification to the parton structure of the bound nucleon still represents a fascinating puzzle to solve. Promising insights in this respect are offered by a new generation of semi-inclusive and exclusive experiments, performed in particular at Jefferson Lab (JLab). This kind of measurements could be able to give new hints into the problem as shown in Ref. [2,3]. Among exclusive processes, a powerful tool is Deeply Virtual Compton Scattering (DVCS). In DVCS, the QCD content of the target is described through non-perturbative functions, the so-called generalized parton distributions (GPDs), which provide a wealth of novel information (for an exhaustive report, see, e.g., Refs. [4], [5]). In particular, GPDs allow to achieve a 3-dimensional view of the inner parton content in the coordinate space. In this talk, we will show the possiblity to obtain a parton tomography of the target [6], either nucleus or nucleon. In facts, in a nucleus, such a DVCS process can happen through two different channels: the coherent one, where the nucleus remains intact and the tomography of the whole nucleus can be done, and the incoherent one, where the nucleus breaks up, one proton is detected and its structure can be accessed. A golden target for this kind of studies is represented by the $^4$He nucleus. In facts, being the lightest system showing the dynamical features of a typical atomic nucleus, it is a paradigmatic system to keep under scrutiny. Moreover, it is scalar and isoscalar and its description in terms of GPDs is easy. This feature makes the $^4$He nucleus a golden target also from the experimental point of view. In facts, recently, DVCS data for this target have become available at Jefferson Laboratory (JLab) where the two DVCS channels have been separated, for the first time [7,8]. These data and the accuracy of the forth-coming ones require rigorous and up-to-date models to be proper interpreted. Previous calculations for $^4$He have been performed long time ago [9–12], in some cases in kinematical regions different from those probed at JLab. We propose a workable approach where conventional nuclear physics effects, described in terms of realistic wave functions, could be properly evaluated and not mistaken for exotic ones. Such a kind of realistic calculations, although very challenging, are possible for a few-body system (e.g. see Ref. [13] for $^2$H and Ref. [14–16] for $^3$He) as the target under scrutiny. In this talk, a review of our main results obtained from the study of the handbag contribution to both DVCS channels, in Impulse Approximation (IA), is presented.

## 2   General DVCS formalism

In this section, the general formalism for both DVCS channels, whose handbag approximation will be studied in IA, is presented. In this scenario, we assume that the interaction of the virtual photon occurs with one quark in one nucleon in $^4$He. Then, the quark is reabsorbed by the target with a transfer of momentum and a real photon is emitted and detected. In IA, only nucleonic degrees of freedom are considered and any further possible interactions of the struck proton with the remnant systems is considered. In other words, possible effects due to final state interaction (FSI) are disregarded. As a reference frame, we choose that where the target is at rest and $\phi$ is the angle between the leptonic and hadronic planes. In this frame, the angle $\phi$ corresponds to the azimuth the outgoing proton. Defining $p(p')$ the initial (final) momenta for initial nuclear (in the coherent channel)/nucleonic (in the incoherent channel) system and analougsly $q_1(q_2)$ for the photons, the experimental variables describing the process are the Bjorken variable $x_B$, $Q^2 = -q_1^2 = -(k-k')^2$, $\Delta^2 = (p'-p)^2 = (q_1-q_2)^2$ and $\phi$. For this kind of process, if the initial photon virtuality $Q^2$ is much larger than the momentum transferred to the hadronic system, the factorization property allows to distinguish the hard vertex, fully known in a perturbative way, from the soft part, where our ignorance about the inner content of the target is encoded. This part is parametrized in terms of GPDs. These

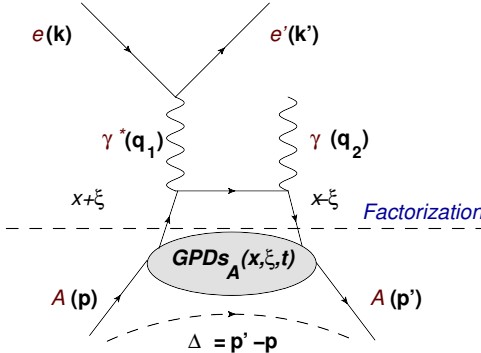

Figure 1: Handbag approximation to the coherent DVCS off a nuclear target.

objects, besides $Q^2$ and $\Delta^2$, depend also on the so-called skewness $\xi = -\frac{\Delta^+}{(p+p')^+}$ [1] i.e., the difference in plus momentum fraction between the initial and the final states, and on $x$, the average plus momentum fraction of the struck parton with respect to the total momentum. Since $x$ is not experimentally accessible, GPDs cannot be directly measured. For this reason, it is useful introducing the so called Compton Form Factors (CFFs) related to GPDs ($H_q$) in the following way ($e_q$ is the quark electric charge, i.e $q = u, d, s$):

$$\Im m \mathcal{H}(\xi, t) = \sum_q e_q^2 \left[ H_q(\xi, \xi, \Delta^2) - H_q(-\xi, \xi, \Delta^2) \right], \tag{1}$$

$$\Re e \mathcal{H}(\xi, t) = \Pr \sum_q e_q^2 \int_0^1 \left( \frac{1}{\xi - x} - \frac{1}{\xi + x} \right) \left[ H_q(x, \xi, t) - H_q(-x, \xi, t) \right]. \tag{2}$$

In this way, CFFs are observable and the experimental way to access these quantities is the beam spin asymmetry (BSA), that for the target under scrutiny, unpolarized (U) by definition, is given by

$$A_{LU} = \frac{d\sigma^+ - d\sigma^-}{d\sigma^+ + d\sigma^-}, \tag{3}$$

where, thanks to different beam spin polarization of the electron beam (L), the differential cross section for $L = \pm$ appears. Since $A_{LU}$ is the observable recently tested at JLab (see Refs. [7, 8]) a realistic calculation of conventional nuclear effects corresponding to a plane wave impulse approximation analysis has been developed and presented in the following.

## 3 Coherent DVCS off $^4$He

The most general coherent DVCS process $A(e, e'\gamma)A$ shown in Fig.1 allows to study the partonic structure of the recoiling whole nucleus $A$ through the formalism of GPDs. In the IA scenario presented above, a workable expression for $H_q^{^4He}(x, \xi, \Delta^2)$, the GPD of the quark of flavor q in the $^4$He nucleus, is obtained as a convolution between the GPDs $H_q^N$ of the quark of flavor $q$ in the bound nucleon N and the off-diagonal light-cone momentum distribution of N in $^4$He and reads

$$H_q^{^4He}(x, \xi, \Delta^2) = \sum_N \int_{|x|}^1 \frac{dz}{z} h_N^{^4He}(z, \xi, \Delta^2) H_q^N \left( \frac{x}{\zeta}, \frac{\xi}{\zeta}, \Delta^2 \right). \tag{4}$$

---

[1] We adopt the notation $a^{\pm} = \frac{a_0 \pm a_3}{\sqrt{2}}$.

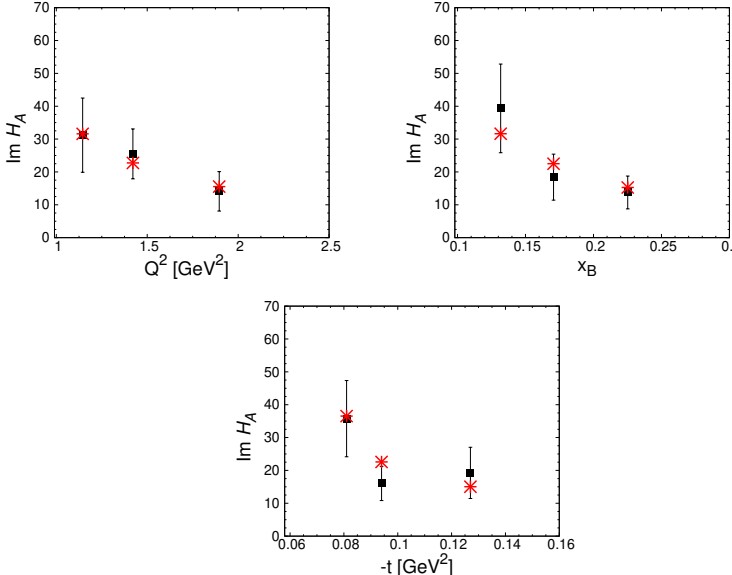

Figure 2: Immaginary part of CFFs given by Eq. (1) obtained from our model for $H_q^{4He}$. Our results (red stars) compared with data (black squares) [7]. From left to right, the quantity is shown in the experimental $Q^2$, $x_B$ and $t = \Delta^2$ bins, respectively. Analogous results for the real part of CFFs given by Eq. (2) are presented in [24].

The light cone momentum distribution appearing in the previous equation is defined as

$$h_N^{4He}(z, \Delta^2, \xi) = \int dE \int d\vec{p}\, P_N^{4He}(\vec{p}, \vec{p} + \vec{\Delta}, E)\delta\left(z - \frac{\bar{p}^+}{\bar{P}^+}\right), \qquad (5)$$

where $P_N^{4He}(\vec{p}, \vec{p} + \vec{\Delta}, E)$ is the off diagonal spectral function. In general,the spectral function is a very complicated object; here, it has the additional feature of being non diagonal. In facts the diagonal spectral function $P_N^{4He}(\vec{p}, E)$ represents the probability amplitude to have a nucleon leaving the nucleus with momentum $\vec{p}$ and leaving the recoiling system with an excitation energy $E^* = E - |E_A| + |E_{A-1}|$, with $|E_A|$ and $|E_{A-1}|$ the nuclear binding energies. Additionally, the off diagonal spectral function $P_N^{4He}(\vec{p}, \vec{p} + \Delta, E)$ accounts for a nucleon reabsorbed by the nucleus with a momentum transfer $\vec{\Delta}$. A complete evaluation of $P_N^{4He}$ should account for all the possible intermediate states of one nucleon and the $A - 1$ body spectator system that can be both a bound (i.e. $E^* = 0$) or a continuum state (i.e. $E^* \neq 0$). Thus, a full realistic evaluation of such an object requires an exact description of all the $^4$He spectrum; for this reason, it represents a challenging few body problem,for which only early attempts exist [17, 18]. So, while the complete evaluation of this object has just begun, as an intermediate step in the present calculation a model of the nuclear non-diagonal spectral function based on the diagonal one proposed in Ref. [19], based on the momentum distribution corresponding to the Av18 NN interaction Ref. [20] and including 3-body forces [21], has been used. Our model used in the present approach can be sketched as follows:

$$\begin{aligned}
P_N^{4He}(\vec{p}, \vec{p} + \vec{\Delta}, E) &= n_0(\vec{p}, \vec{p} + \vec{\Delta})\delta(E) + P_1(\vec{p}, \vec{p} + \vec{\Delta}, E) \\
&\simeq a_0(|\vec{p}|)a_0(|\vec{p} + \vec{\Delta}|)\delta(E) + \sqrt{n_1(|\vec{p}|)n_1(|\vec{p} + \vec{\Delta}|)}\delta(E - \bar{E}) \qquad (6)
\end{aligned}$$

where we made use of the momentum distribution $n(|\vec{p}|) = n_0(|\vec{p}|) + n_1(|\vec{p}|)$. In particular, when the recoiling system is in its ground state, the momentum distribution $n_0(|\vec{p}|)$ is realistically evaluated along the scheme of Ref. [22] in terms of exact wave functions of 3- and

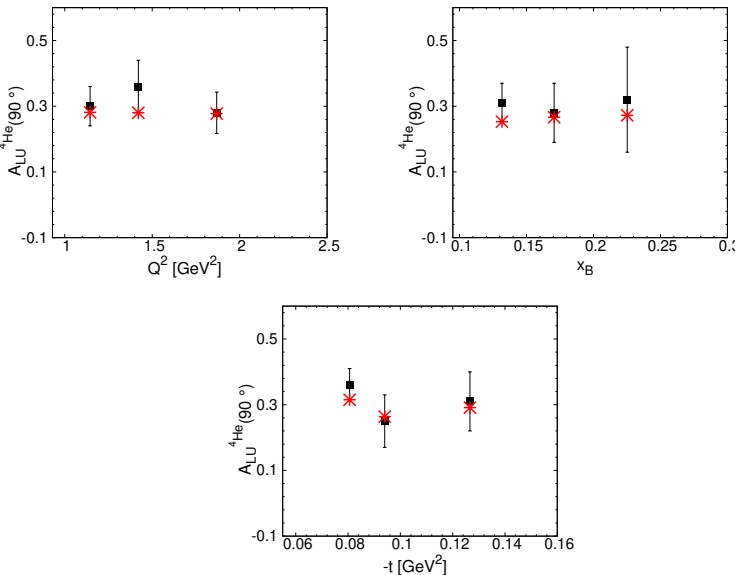

Figure 3: $^4$He azimuthal beam-spin asymmetry $A_{LU}(\phi = 90^o)$ given by Eq. (8): results of Ref. [24] (red stars) compared with data (black squares) [7]. From left to right, the quantity is shown in the experimental $Q^2$, $x_B$ and $t = \Delta^2$ bins, respectively

4-body systems, i. e.

$$n_0(|\vec{p}|) = |<\Phi_3(1,2,3)\chi_4\eta_4|j_0(|\vec{p}|R_{123,4})\Phi_4(1,2,3,4)>|^2. \tag{7}$$

As far the excited part $n_1(|\vec{p}|)$ concerns, it has been obtained starting from the total momentum distribution calculated in terms of the non diagonal density matrix obtained, again, from the realistic wave function of the $^4$He nucleus. In our model, both the angular and the energy dependence are modelled. In particular, in the excited sector, the energy is fixed to an average value for the recoiling system chosen so that the non diagonal spectral function reduces to the diagonal one (see Ref. [19]).

Concerning the nucleonic GPD appearing in Eq.(4), the well known GPD model of Ref. [23] has been used. We remind that, in principle, the model is valid for $Q^2 \geq 4$ GeV$^2$.

With these ingredients at hand, as an encouraging check, typical results are found, in the proper limits, for the nuclear charge form factor and for nuclear parton distributions. A complete explanation and relevant plots can be found in Ref. [24]. With our model for $H_q^{4He}$, a numerical comparison with the Eqs. (1) and (2), experimentally accessed by JLab could be done. Finally, a comparison with BSA of the coherent DVCS channel containing the previous quantities and reading

$$A_{LU}(\phi) = \frac{\alpha_0(\phi)\,\Im m(\mathcal{H}_A)}{\alpha_1(\phi) + \alpha_2(\phi)\,\Re e(\mathcal{H}_A) + \alpha_3(\phi)\left(\Re e(\mathcal{H}_A)^2 + \Im m(\mathcal{H}_A)^2\right)} \tag{8}$$

has been done. Here above, $\alpha_i(\phi)$ are kinematical coefficients defined in Ref. [25]. The comparison between our results and the experimental data shown in Fig. 3 are satisfactory [24]. One can conclude that a careful analysis of the reaction mechanism in terms of basic conventional ingredients is successful and that the present experimental accuracy does not require the use of exotic arguments, such as dynamical off-shellness.

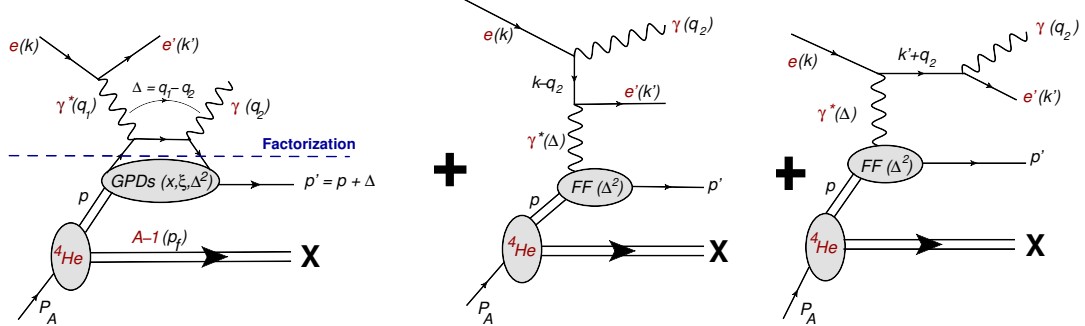

Figure 4: Incoherent DVCS off $^4$He in IA. To the left, pure DVCS contribution; to the right the two Bethe Heitler terms.

# 4 Incoherent DVCS off $^4$He

In the process $A(e, e'\gamma p)X$ depicted in Fig. 4, fascinating insights about the parton structure of the bound proton can be accessed. This new information, compared with those known for the free nucleon could provide a pictorial view of the realization of the EMC effect [1]. In order to have a complete evaluation of Eq. (3), the explicit expression for the cross-section for a DVCS process occurring off a bound moving proton in $^4$He is required. Working within an IA approach, we account for the pure kinematical off-shellness of the initial bound proton obtaining a convolution formula for the cross sections describing the process that reads

$$d\sigma_{Inc}^{\pm} = \int_{exp} dE\, d\vec{p}\, \frac{p \cdot k}{p_0\, |\vec{k}|} P^{^4He}(\vec{p}, E)\, d\sigma_b^{\pm}(\vec{p}, E, K). \tag{9}$$

If one differentiates the previous expression with respect to the experimental variables, the following equation for different beam polarization, explicitly appearing in the expression of BSA (3), reads

$$d\sigma^{\pm} \equiv \frac{d\sigma_{Inc}^{\pm}}{dx_B dQ^2 d\Delta^2 d\phi} = \int_{exp} dE\, d\vec{p}\, P^{^4He}(\vec{p}, E)|\mathcal{A}^{\pm}(\vec{p}, E, K)|^2 g(\vec{p}, E, K),$$

where $K$ is the set of kinematical variables $\{x_B, Q^2, t, \phi\}$. The range of these variables probed in the experiment selects only the relevant part of the diagonal spectral function $P_N^{^4He}(\vec{p}, E)$, which has therefore to be integrated in a restricted region range called $exp$. The quantity $g(\vec{p}, E, K)$ is a complicated function arising from the integration over the phase space and including also the flux factor $p \cdot k/(p_0\, |\vec{k}|)$. In the above equation, the squared amplitude includes three different terms, i.e $\mathcal{A}^2 = T_{DVCS}^2 + T_{BH}^2 + \mathcal{I}_{DVCS-BH}$ as shown in Fig. 4 and each contribution has to be evaluated for an initially moving proton. Our amplitudes generalize the ones obtained for a proton at rest in Ref. [26] and the main assumptions done are summarized in Ref. [27]. Since in the kinematical region of interest at Jlab the Bethe Heitler (BH) part is dominating, the key partonic insights are all hidden in the numerator of the BSA that selects the interference DVCS-BH. In this way, the asymmetry reads

$$A_{LU}^{Incoh}(K) = \frac{\int_{exp} dE\, d\vec{p}\, P^{^4He}(\vec{p}, E)\, g(\vec{p}, E, K)\, \mathcal{I}_{DVCS-BH}(\vec{p}, E, K)}{\int_{exp} dE\, d\vec{p}\, P^{^4He}(\vec{p}, E)\, g(\vec{p}, E, K)\, T_{BH}^2(\vec{p}, E, K)}. \tag{10}$$

Since our ultimate goal is to have a comparison with the experimental BSA, that actually is a function of the angle $\phi$ of the outgoing proton, we exploit the azimuthal dependence of Eq.(10)

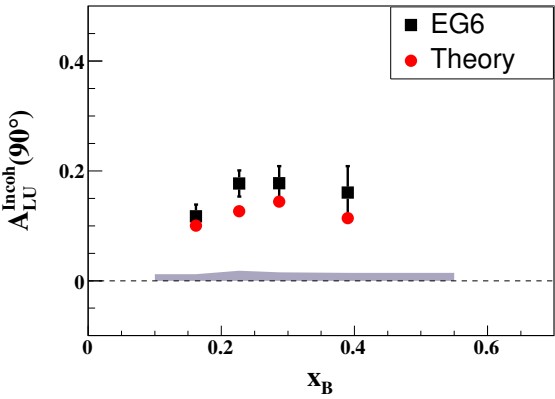

Figure 5: Azimuthal beam-spin asymmetry for the proton in $^4$He, $A_{LU}^{Incoh}$, Eq.(10), for $\phi = 90^o$: results of this approach [27](red dots) compared with data (black squares) [8]

.

decomposing in $\phi$ harmonics the interference and the Bethe Heitler parts. All the information about the parton content of the bound proton is encapsulated in the imaginary part of CFF containing the GPDs. In our model, the modification to the inner content of the bound proton is accounted by the rescaling of the skewness $\xi'$, that depends explicitly on the 4-momentum components of the initial proton. In the present calculation we considered only the dominating contribution given by the $H_q(x, \xi', t)$ GPD, for which use of the GK model has been made [23]. Concerning the diagonal spectral function, we made use of the model presented in Ref. [19]. The ground contribution is evaluated considering the realistic momentum distribution given by Eq. (7) while the excited part is an update of the model presented in Ref. [28] which considers 2N correlations. The results are depicted in Fig. 5 (See details and analogous plots in [27]). As expected, the agreement with experimental data is good except the region of lowest $Q^2$, corresponding to the first $x_B$ bin. In this region, in facts, the impulse approximation is not supposed to work well, since final state interaction effects, otherwise neglected in IA, could be sizable. This fact requires a careful evaluation of the interplay between $\Delta^2$ and $Q^2$ as already notices in Ref. [27]. An interesting quantity to study in order to appreciate the nuclear effects foreseen by our model is the ratio between the asymmetry for an off-shell bound proton and the corresponding quantity for the free proton. In this way, it would be possible to study whether the difference observed in Ref. [8] is linked to a modification of the inner structure of the proton related to the EMC effect or to another nuclear effect. This ratio and its meaning is deeply discussed in Ref. [27].

## 5   Conclusions

We can conclude that for both DVCS channels, considering the present experimental accuracy, the description of the data does not need the use of exotic arguments, such as dynamical off shellness.

An improved treatment of both the nucleonic and the nuclear parts of the evaluation is needed for a serious benchmark calculation in the kinematics of the next generation of precise measurements at high luminosity [29]. The latter task includes the computation of realistic computation of a one-body non diagonal (for the coherent channel) and diagonal (for the incoherent channel) $^4$He spectral function. Work is in progress towards this challenging purpose.

In the meantime, the straightforward approach summarized in this talk represents a workable framework for the planning of the next measurements.

## Acknowledgements

This work was supported in part by the STRONG-2020 project of the European Union's Horizon 2020 research and innovation programme under grant agreement No 824093, and by the project "Deeply Virtual Compton Scattering off $^4$He", in the programme FRB of the University of Perugia.

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
