# Peer review of "Theoretical study of deeply virtual Compton Scattering off $^4$He"

_SciPost Physics Proceedings_

## Round 1 · Referee Report · Anonymous (Referee 1) · 2019-10-16

Strengths

  1. Reports on a study of DVCS, which is a hot topic on both theoretical and experimental fronts.
  2. Presents an approach which is feasible and workable, yet realistic and rigorous enough by virtue of properly evaluating nuclear physics effects involving appropriate nuclear wave-functions, and by using a complex enough approximation of the spectral function (non-diagonal for the coherent and diagonal for the incoherent channel).
  3. Shows (within uncertainties) that "exotic" mechanisms, such as dynamical off-shellness, are not needed to describe the existing data.

Weaknesses

None except for linguistic issues (see below).

Report

The report focuses on the DVCS process on He4, a scalar/isoscalar nucleus ideal for both experimental and theoretical studies. The motivational background is the description of the process dynamics in terms of the GPDs and the corresponding 3d mapping of the target's parton content. The Authors were able to exploit the Impulse Approximation to assess the validity of the handbag diagram for the coherent process and the sufficiency of the DVCS-Bethe-Heitler interference to investigate the incoherent process, and found remarkable agreement with the recent experimental results from JLab. The contribution, therefore, is a valuable addition to our understanding of the DVCS processes at JLab energies, and is a welcome ingredient in planning future high-luminostity, high-precision experiments. This contribution certainly deserves to be published in these Proceedings.

Requested changes

Spell check.
Fix typos (for example "In facts" -> "In fact", several times; line 6 of Sec. 2: "considered" -> "neglected"?, etc.)

---

## Editorial Decision

resubmitted